# Incorporating Aminated Nanodiamonds to Improve the Mechanical Properties of 3D-Printed Resin-Based Biomedical Appliances

**DOI:** 10.3390/nano10050827

**Published:** 2020-04-26

**Authors:** Utkarsh Mangal, Ji-Young Seo, Jaehun Yu, Jae-Sung Kwon, Sung-Hwan Choi

**Affiliations:** 1Department of Orthodontics, Institute of Craniofacial Deformity, Yonsei University College of Dentistry, 50-1 Yonsei-ro, Seodaemun-gu, Seoul 03722, Korea; utkmangal@yuhs.ac (U.M.); jyseo13@yuhs.ac (J.-Y.S.); hun718@yuhs.ac (J.Y.); 2BK21 PLUS Project, Yonsei University College of Dentistry, Seoul 03722, Korea; 3Department and Research Institute of Dental Biomaterials and Bioengineering, Yonsei University College of Dentistry, Seoul 03722, Korea

**Keywords:** additive manufacturing, aminated nanodiamond, poly (methyl methacrylate), nanocomposites, mechanical properties

## Abstract

The creation of clinically patient-specific 3D-printed biomedical appliances that can withstand the physical stresses of the complex biological environment is an important objective. To that end, this study aimed to evaluate the efficacy of aminated nanodiamonds (A-NDs) as nanofillers in biological-grade acrylate-based 3D-printed materials. Solution-based mixing was used to incorporate 0.1 wt% purified nanodiamond (NDs) and A-NDs into UV-polymerized poly(methyl methacrylate) (PMMA). The ND and A-ND nanocomposites showed significantly lower water contact angles (*p* < 0.001) and solubilities (*p* < 0.05) compared to those of the control. Both nanocomposites showed markedly improved mechanical properties, with the A-ND-containing nanocomposite showing a statistically significant increase in the flexural strength (*p* < 0.001), elastic modulus (*p* < 0.01), and impact strength (*p* < 0.001) compared to the control and ND-containing groups. The Vickers hardness and wear-resistance values of the A-ND-incorporated material were significantly higher (*p* < 0.001) than those of the control and were comparable to the values observed for the ND-containing group. In addition, trueness analysis was used to verify that 3D-printed orthodontic brackets prepared with the A-ND- and ND-nanocomposites exhibited no significant differences in accuracy. Hence, we conclude that the successful incorporation of 0.1 wt% A-ND in UV-polymerized PMMA resin significantly improves the mechanical properties of the resin for the additive manufacturing of precisive 3D-printed biomedical appliances.

## 1. Introduction

Technological advancements in recent years have driven product design and manufacturing processes. Three-dimensional (3D) printing or direct digital manufacturing in the form of additive manufacturing has offered unique advantages for the manufacturing of intricate custom designs with short lead-times and small environmental footprints [1]. Improved accessibility has led to the rapid adaptation of these techniques in the field of biomaterials and has paved the way for the practice of precision treatment designs [2].

Poly (methyl methacrylate)-based (PMMA-based) acrylic resins are a commonly used material in both medical and dental applications because of their cost effectiveness, appropriate mechanical strengths, and ease of handling. PMMA-based polymeric materials also form a representative group of materials for the biomedical additive manufacturing of patient-specific and sophisticated biomedical appliances (e.g., cranial implants, maxillofacial surgical splints, intraoral appliances and as biomimetic scaffolds) [3]. However, the physical properties of most resin-based appliances deteriorate as a result of the nature of their constituent materials, which absorb fluids, such as saliva, in the harsh oral environment, leading to a reduction in long-term performance [4].

Noteworthy progress toward improving polymer properties has been made through the application of carbon-based nanofillers (CBNs), such as carbon nanotubes, graphene, and nanodiamonds (NDs) [5,6]. Among these CBNs, NDs present properties that mimic the features of their bulk form, including superior mechanical strength and extreme surface hardness [7]. Among various CBNs, NDs exhibit a rich surface chemistry and superior biocompatibility, which are added advantages of their use [8]. The above characteristics make low-cost synthesized NDs an appropriate choice as fillers for the production of enhanced polymer nanocomposites for biomedical applications [9].

Several studies of polymer–ND composites have shown that the use of NDs can improve the mechanical properties of polymer matrices [5]. However, the majority of these studies have focused on epoxy-, poly (vinyl alcohol)-, or poly(lactic acid)-based materials. Presently, only a few studies have evaluated the efficacy of NDs as fillers in PMMA-based polymers for biomedical use [9,10]. Moreover, the majority of these studies deployed typical manual casting methods to directly fabricate suspensions of NDs in polymers with mechanical stirring alone. Although the results are encouraging, the use of traditional techniques is limited with respect to fabrication time, design complexity, and labor intensiveness, compared to additive-manufacturing methods.

Furthermore, good ND homogeneity in the polymer matrix is essential for ensuring consistent improvements through the incorporation of NDs. In addition to good dispersion, factors such as the nature of the nanofiller/polymer interface are critical for improving the properties [11]. Pristine NDs present a variety of oxygen-rich functional moieties that enable the surface chemistry of the ND to be tailored in order to expand its properties. Among the various moieties functionalized onto an ND surface, two functional groups that are stable in biomedical research applications are the carboxylic acid (-COOH) and amino (-NH_2_) groups. In recent years, various research groups have attempted to incorporate aminated nanodiamonds (A-NDs) into polymers in order to enhance their mechanical properties, with varying degree of success reported [5,8]. Polymers of lactic acid [12] and epoxides [13] have experienced definite improvements with A-NDs as nanofillers, although acrylate-based polymers have hardly been explored, especially acrylate-based 3D printed materials.

Taking this into account, in the present study, we evaluated the efficacy of A-NDs as nanofiller in biological-grade acrylate-based 3D printed materials, with the aim of creating patient-specific 3D-printed biomedical appliances that can withstand the physical stresses of the complex biological environment. The following null hypothesis was considered: there will be no differences in properties between PMMA, PMMA incorporated with NDs, and PMMA incorporated with A-NDs, including the quality of additive-manufactured samples.

## 2. Materials and Methods

### 2.1. Materials

A commercially available UV-curable acrylate-based resin material intended for the production of orthodontic customized appliances (NextDent Ortho Rigid, 3D Systems, NextDent B.V., Soesterberg, The Netherlands) was used in all the experiments. The Ortho Rigid resin is a viscous liquid (0.8 to 1.5 Pa) having a relative density (water) of 1.11 to 1.15, showing better solubility in organic solvents than water. Chemically, the resin is composed of methacrylic oligomers (>90% *w/w*) with a minor percentage of phosphine oxides (<3% *w/w*) having UV-sensitive (Blue UV-A 315–400) initiators for polymerization.

Nanodiamond powder (N0962) and anhydrous aminated (-NH_2_) nanodiamond (N0968) powder were purchased from Tokyo Chemical Industry (TCI Co., Ltd. (Tokyo, Japan)), both with purities greater than 97%. The as-received ND powder was gray in color, with an average particle size of less than 10 nm, according to the manufacturer.

### 2.2. Synthesis of Purified and Aminated ND-Incorporated Nanocomposites

The purified ND-incorporated and A-ND-incorporated nanocomposites were prepared using purified ND and A-ND, respectively, at 0.1 wt% of the neat resin. The non-functionalized ND powder was purified to remove non-diamond impurities following the previously described procedure adopted in an earlier publication [14]. The A-ND powder was used as supplied.

The nanoparticles were dispersed at the nano-level in the polymer matrix by solvent-based mixing using chloroform (CHCl_3_, Sigma-Aldrich, St. Louis, MO, USA) [15]. The purified or A-ND suspension in chloroform was stirred at 50 °C for 30 min and then probe-sonicated using with a 50–60 Hz sonicator (Q125, Qsonica, LLC., Newtown, CT, USA). The neat resin was magnetically stirred at 60 °C for 30 min to lower its viscosity. To promote homogenous dispersion, the suspension was then mixed with a predetermined weight of the low viscosity resin and magnetically stirred for 24 h at 60 °C, after which the solvent was slowly evaporated over 2 d with continuous stirring (500 rpm) at 60 °C. The resultant ND-resin suspension was degassed under vacuum for 2 h and used as is without any further modifications (Figure 1).

All specimens were fabricated using a digital light processing (DLP) 3D printer (NextDent 5100, 3D Systems, NextDent B.V., Soesterberg, The Netherlands) with the stereolithography data for each specimen prepared according to the measurement method using 3D design and modelling software (Autodesk^®^ Meshmixer, Autodesk, San Rafael, CA, USA). The specimens were polymerized under 405-nm light and post-processed according to the manufacturer’s instructions using a UV oven (NextDent LC-3DPrint Box, 3D Systems, NextDent B.V., Soesterberg, The Netherlands).

### 2.3. Characterizing the ND- and A-ND-Incorporated Nanocomposites

Transmission electron microscopy (TEM; JEM-200F, JEOL, Tokyo, Japan) was used to qualitatively characterize the aggregation tendency of the NDs and A-NDs in resin. To that end, 0.1 mg of each sample was dispersed in 1 mL of ethanol, and one drop of the suspension was deposited on a TEM grid (200 mesh) and dried under vacuum. Images were acquired at 200 kV. To quantitatively compare the aggregation tendency of the NDs and A-NDs, 1 mg/mL of each sample was measured by dynamic light scattering with Zetasizer Nano ZS90 (Malvern Instruments Ltd., Malvern, United Kingdom).

The control and the ND- and A-ND-incorporated samples, along with the ND nanofillers themselves, were subjected to Fourier-transform infrared (FT-IR) spectroscopy (Nicolet™ iS™ 10 FTIR spectrometer, Thermo Fisher Scientific, Korea). Each sample was washed twice with ethanol to remove impurities prior to the acquisition of its spectrum.

For thermogravimetric analysis (TGA), a TG/DTA 7300 (Seiko Instruments Inc., Chiba, Japan) was employed. TGA was performed on the ND and A-ND nanodiamond powder, the control and ND- and A-ND-incorporated samples, in the presence of nitrogen. The equilibration at 100 °C for 20 min was followed by a ramp of 10 °C/min up to 800 °C. A total of 2–5 mg of each sample was weighed and placed in an alumina crucible for analysis.

The nano-polymerized specimens were characterized by examining their surface morphologies and fracture-surface patterns created with a computer-controlled universal testing machine at a crosshead speed of 5 mm/min. Field-emission scanning electron microscopy (FE-SEM) images were acquired on a JEOL-7800F microscope (Tokyo, Japan). All samples were coated with 5-nm-thick Pt using an ion coater (ACE600, Leica, Wetzlar, Germany).

### 2.4. Water Contact Angle and Hydrophilicity

The contact angles of the samples (n = 5) were measured to evaluate the hydrophobicities/hydrophilicities of the ND- and A-ND-incorporated nanocomposites. The contact angle (θ) was determined by measuring the angle between the examined flat surface and the tangent created by a drop of dH_2_O at the point of contact with the surface. To that end, dH_2_O (3 µL) was placed at 1.0 µL·s^−1^ at the center of each sample surface at room temperature. Contact angles were measured using a droplet analysis device (SmartDrop, Femtofab, Seongnam-si, Korea) and the sessile-drop method.

### 2.5. Water Sorption and Solubility

Disc-shaped specimens (d = 15.0 mm, h = 1.0 mm, n = 5) were prepared for each group in accordance with a previously reported method [16]. The specimens were kept in a desiccator maintained at 37 ± 2 °C for 22 h, after which they were transferred into another desiccator maintained at 23 ± 1 °C for 2 h and weighed to an accuracy of ± 0.1 mg. The initial mass (m1) was recorded when the mass of the specimen did not change by more than 0.1 mg in any 24 h period. After final drying, the mean diameter and thickness of each specimen was measured to an accuracy of ±0.01 mm and the volume of the sample was calculated (V; mm^3^). Each specimen was immersed in 15 mL of distilled water at 37 ± 1 °C for 7 d. The surface water was removed by blotting to eliminate any visible moisture, after which each specimen was waved about in the air for 15 s and weighed (m2) within 60 s of its removal from the water. The specimens were then stored in desiccators as described above until a constant mass (m3) was achieved. The water sorption (Wsp = (m2 − m3)/V) and the water solubility (Wsl = (m1 − m3)/V) values were then calculated.

### 2.6. Flexural Strength and Modulus

In the present study, mechanical properties were analyzed following the ISO 20795-2 International Standard, which is related to the specific applications of the PMMA of interest in this study, namely orthodontic appliances. Eighteen samples (n = 6, control, ND-, and A-ND-incorporated) were printed with average dimensions of 3.3 mm × 10 mm × 64 mm. A computer-controlled universal testing machine (Model 3366, Instron, Norwood, MA, USA) was used to fracture the specimens through three-point flexuring with a 1-kN load cell. The flexural strength (σ_f_) and elastic modulus (E_f_) were measured at a span length of 50 mm and a crosshead speed of 5 mm/min. The flexural strength and elastic modulus were calculated using standard equations. In addition, the modulus of resilience (Ur) was calculated as: *Ur = σ_f_^2^⁄2E_f_*.

### 2.7. Impact Strength

Impact strength was determined by the IZOD impact test according to a modified ASTM D256 specification. Ten unnotched rectangular specimens (l = 64 mm, h = 12.7, and b = 3.2 mm) were printed. The specimens were then subjected to impact testing using an IZOD impact apparatus (Yasuda Seiki Seisakusho Ltd., Tokyo, Japan) with a swinging pendulum (0.598 g, 0.33 m). Impact strength is expressed as energy per unit and calculated using the formula: IS = E/hb, where E is the absorbed energy calculated as the difference between the nominal and dynamic energy of the pendulum after specimen fracture.

### 2.8. Surface Hardness and Wear Resistance

A Vickers hardness machine (DMH-2, Matsuzawa Siki Co. Ltd., Akita, Japan) was used to determine sample hardness by applying a force of 300 gf (2.94 N) for 30 s; an average value was calculated from three different locations on each specimen (n = 5).

Surface wear resistance was determined using a previously reported method [17]. Three samples from each group were polished with increasing fineness using SiC paper (800, 1200, and 1500 grit, sequentially). Following initial surface-roughness analysis, a V8 cross-brushing machine (Sabri Co., Downers Grove, IL, USA) was used to simulate toothbrushing with a dentifrice slurry at the recommended dilution of 25 g in 40 mL of water. The samples were subjected to abrasion with a mechanized toothbrush at 200 g load, with a brushing velocity of 50 times/min for 5000 strokes, which simulates 3–4 months of tooth brushing [18]. Average surface roughness values were recorded before and after toothbrush wear, using a 3D optical profilometer (ContourGT-X 3D Optical Profiler, Bruker, Billerica, MA, USA) in non-contact mode. Representative 3D images and corresponding surface roughness values (Ra) were recorded at three different points (at the center and 2-mm intervals on either side) and averaged.

### 2.9. Hydro-Thermal Fatigue Testing

To simulate accelerated physiological aging of the nanopolymers, the commonly employed method of thermocycling was used [19]. Specimens with same design as in mechanical testing were prepared and stored dry at 23 ± 2 °C for 24 ± 2 h. Since there is no definite standardization for bath temperatures or for number of cycles for thermocycling, the recurrently used method advocated by the International Organization for Standardization (ISO)/TS 10477:2018 for testing dental materials was used. All thermocycles were conducted between 5 °C and 55 °C for a dwell time of 30 s. Time of filling and emptying the vessel with a working liquid (water) was 15 s for a total of 5000 cycles (RB 508, Thermal Cyclic Tester, R&B Inc, Daejeon, Korea).

### 2.10. Trueness

Fifteen orthodontic-bracket-shaped specimens (n = 5, control, ND-, and A-ND-incorporated) were fabricated using a 3D printer (NextDent 5100, 3D Systems, NextDent B.V., Soesterberg, the Netherlands). The samples were then scanned using a 3Shape E3 scanner (3Shape, Copenhagen, Denmark). Dimensional accuracies were evaluated for the three groups against a reference computer-aided-design file used for printing brackets. The best-fit superimposition method using a 3D morphometric program (Geomagic^®^ Control X™, 3D Systems, Rock Hill, SC, USA) was used to determine root-mean-square (RMS) values, which indicate trueness between samples [20]. Overall deviations were shown on a color map for intuitive comparison with deviations of ±200 µm and tolerances of ±10 µm assigned.

### 2.11. Statistical Analysis

All statistical analyses were performed using IBM SPSS software, version 23.0 (IBM Korea Inc., Seoul, Korea) for Windows, with data from at least three independent experiments. The results obtained from the control and experimental groups were analyzed by one-way analysis of variance (ANOVA) followed by Tukey’s test. *p* < 0.05 was considered to be statistically significant.

## 3. Results

### 3.1. Characterizing ND- and A-ND-Incorporated Nanocomposites

TEM images of the ND- and A-ND-containing resins show particles that are 4–6 nm in size and are aggregated in both materials; however, the NDs appear to be more aggregated than the A-NDs. The particles in the latter are somewhat more homogeneously dispersed, as is evident in Figure 2A.

To observe the aggregation tendency and cluster size distribution, particle size analysis was conducted. The results from dynamic light scattering showed marked differences between the NDs and A-NDs, as appreciated in the histograms in Figure 2B. The NDs showed an aggregation tendency, with sizes in excess of 100 nm having 62.1% of aggregates in size ranges of 160–255 nm and 8.8% of aggregates greater than 1 µm in size. In contrast, with A-NDs, 90% of the aggregates were found in the size range of 20–40 nm.

The FT-IR spectra of the nanofiller powders and polymerized specimens are displayed in Figure 3A,B, which reveals the presence of bands at 2815–2964 cm^−1^ (C-H stretching), 3448 cm^−1^ and 1554 cm^−1^ (N-H stretching and bending), and 1632 cm^−1^ (C=C) in the spectrum of the A-ND powder. The comparison of the FTIR spectra of powder of ND with A-ND (Figure 3A) shows a distinctive peak at 1554 cm^−1^, which is observed in the IR spectrum of A-ND and absent in the spectrum of ND. This has been identified as the amide II band, a mixture of C-N stretch and N-H vibrations, reported in the range of 1550 ± 20 cm^−1^ [21]. The peak observed at 1630 ± 10 cm^−1^ for both ND and A-ND can be assigned to the bending mode of O-H or N-H bonds.

In the higher frequency range, maximum intensity in the 2800–3500 cm^−1^ range can be observed as the red shift in the A-ND (-NH_2_), described as the result of the overlap of the N-H stretch by Mochalin et al. [22]. The FT-IR spectra of the nanocomposites reveal characteristic bands that correspond to acrylate groups, with the main band (1730 cm^−1^) attributable to carbonyl groups.

As seen in Figure 3C, the TGA curve for both ND and A-ND powder does not show any significant mass loss up to 500 °C, which demonstrates their stable nature for this temperature interval. The TGA curve for the control exhibited a low thermal stability with a sharp drop in the weight loss from 270 °C compared to the ND- and A-ND-incorporated nanocomposites, which showed higher thermal stability. In the control group, 40% of weight loss was recorded at 405 °C, while in the ND- and A-ND-incorporated nanocomposites, the same was observed at 420 °C and 426 °C, respectively.

The smooth appearance of the macroscopically fractured section highlights the brittle nature of the control sample. The SEM images in Figure 4A,D reveal that the fracture surface of the control sample is smooth, with minimal features. The fracture surfaces of the ND- (Figure 4B,E) and A-ND-incorporated samples (Figure 4C,F) show the impact of the nanofiller, as markedly rough surfaces were observed for both nanofiller-containing samples, confirming that high energy is required for crack propagation. Moreover, no microscale agglomeration was observed in the nanocomposite samples, on either their surfaces or fracture-surface cross-sections.

### 3.2. Hydrophilicity, Water Sorption, and Solubility

Wettability evaluated through contact-angle analysis revealed statistically significant differences between the control and nanocomposite groups (*p* < 0.001, Figure 5A). The average value of the contact angle was highest for the control group (73.81° ± 4.9°), followed by the A-ND- (48.22° ± 8.01°) and ND-incorporated (17.32° ± 6.4°) groups.

The water sorption results (Figure 5B) show values that decrease from the control (22.55 ± 3.63 μg/mm^3^) to the ND- (21.55 ± 2.44 μg/mm^3^) and A-ND-incorporated (20.62 ± 0.78 μg/mm^3^) groups; however, these values were not statistically significant. In contrast, the solubility results in Figure 5C show marked differences between the three groups (control: 0.168 ± 0.15 μg/mm^3^; ND-incorporated: 0.024 ± 0.35 μg/mm^3^; A-ND-incorporated: −0.348 ± 0.33 μg/mm^3^); the difference between the control and A-ND-incorporated groups was found to be statistically significant (*p* < 0.05).

### 3.3. Mechanical Properties of the Nanocomposites

The ND-incorporated test groups showed higher flexural and impact strengths compared to those of the control (Figure 6A,B; Table 1). Significant differences were observed for flexural strength and impact strength (A-ND > ND > control; *p* < 0.001) between all groups.

The A-ND-incorporated nanocomposite exhibited an elastic modulus that was significantly higher (*p* < 0.01) compared to those of both the ND-incorporated sample and the control, whose values were comparable and not markedly different (Figure 6C). However, both the ND- and A-ND-incorporated samples exhibited moduli of resilience that were significantly higher (*p* < 0.001) that that of the control, although the values for the ND- and A-ND-incorporated samples were not significantly different (Figure 6D).

### 3.4. Surface Microhardness and Wear Resistance

Surface hardness was characterized through micro-indentation studies, which showed that the addition of a nanofiller resulted in a significant (*p* < 0.001) increase in surface hardness; however, no statistical difference was found between the ND- and A-ND-incorporated materials (Table 1, Figure 7A).

Wear resistance was evaluated using the toothbrush-wear method, which revealed significant (*p* < 0.001) and very high differences in the mean changes in the surface roughness of the control (1.571 ± 0.24 µm) and the ND- (0.123 ± 0.10 µm) and A-ND-incorporated (0.211 ± 0.23 µm) samples, which were statistically similar (Figure 7B,C).

### 3.5. Response to Hydro-Thermal Fatigue

The mechanical properties following exposure to 5000 cycles of hydro-thermal fatigue are summarized in Table 2 and Figure 8. Flexural strength was significantly (*p* < 0.001) lower in control compared to both ND-incorporated and A-ND-incorporated nanocomposites, while the nanocomposite groups were comparable. Elastic modulus was significantly (*p* < 0.05) lower in control when compared to A-ND-incorporated nanocomposites, whereas the ND-incorporated group had no significant difference in relation to both control and A-ND-incorporated nanocomposites. The modulus of resilience also showed a similar trend as before thermocycling; however, the statistically significant increase was observed from control to ND to A-ND (*p* < 0.001).

Nanocomposites also maintained significantly higher surface hardness than the control group after thermocycling. However, statistically significant differences were also observed between the A-ND group compared to the ND group.

### 3.6. Trueness

A morphometric comparison of the three groups verified the absence of any significant morphological changes in printing accuracy due to nanofiller incorporation. The observed mean RMS values for the control (0.0469 ± 0.002 µm), ND-incorporated (0.0475 ± 0.0027 µm), and A-ND-incorporated (0.0460 ± 0.0032 µm) groups were not significantly different (*p* = 0.732) when compared statistically. The results were visually appraised by superimposing the scanned images with the reference CAD file to produce color maps showing deviations in the ±200 µm range (Figure 9).

## 4. Discussion

In fabrication of resin-based nanocomposites, achieving a homogenous dispersion of carbon nanoparticles, such as NDs, is challenging, and methods such as high-power sonication and shear mixing have been suggested [23]. However, the choice of method has been based on the suitability of the polymer material used. The exact composition of the Ortho Rigid resin presently used is proprietary information, but to the best of our knowledge, it consists mainly of methacrylic oligomers (>90 wt%). The surface properties of NDs enable them to be chemically modified and, consequently, form strong covalent bonds in the polymer matrix [24]. However, in the context of polymeric fillers, NDs have been observed to agglomerate, leading to poor dispersion and inadequate polymer/ND interactions as a consequence [5]. Therefore, to best use the features of NDs as fillers for polymers such as PMMA, it is imperative that the nanoparticles are highly uniformly dispersed. However, due to the surface chemistry of ND, deagglomeration beyond the microscale with simple mechanical methods alone is challenging; hence, surface modification is required [25]. Surface functionalization of the ND can aid the deagglomeration process and improve stability in organic media by reducing the surface energy and inter-particle bonding of nanoparticles [26,27].

In the previous study by Kalsoom et al. [28], direct vigorous stirring and sonication of the resin were reported to mix microdiamond particles. Optimization studies for incorporation of carbon nanofiller in resin have also been conducted by Prolongo et al. [15] and Garg et al. [29], with recommendations on the use of sonication and magnetic stirring. In the present study, the conditions were also carefully adapted as per these recommendations. Similar methodology could also be observed in the earlier published work by Feng et al. [6]. However, according to the observations of Suave et al. [30] direct resin sonication can lead to a reduction in oligomer concentration due to the lysing of chains. Hence, in the present study, we used chloroform, one of the most used solvents, to disperse NDs with high-power sonication prior to mechanical mixing and degassing. The other key factors that need to be considered to avoid agglomeration and uneven dispersion are surface homogenization and the concentration of the incorporated NDs. The properties of PMMA [9,10], epoxy [14,31], and poly (vinyl alcohol) [32] were reportedly improved with less than 0.2 wt% ND incorporation. Therefore, in the present study, both the ND- and A-ND-incorporated nanocomposites were fabricated with a 0.1 wt% ND concentration.

Intraoral appliances in orthodontics are prescribed for long continuous periods of wear; consequently, improvements in surface wettability will lead to improvements in retention between oral mucosa and the intaglio surface [33]. In the present study, the surface features of the ND were observed to influence the features of the polymer surface, with markedly significant reductions in water contact angle observed for both the ND- and A-ND-nanocomposite groups, in agreement with the observations of a previous study and highlighting the influence of the hydrophilic nature of diamond [34]. The difference between the ND- and A-ND-incorporated groups, however, can be attributed to changes in the surface chemistry resulting from the addition of amino groups, which reduces the surface reactivity of the oxygenated outer shell compared to that of pure ND, as previously described [35]. In addition, it is plausible that the surface chemistry influences the water solubility of the polymer, with low or negative values observed for the nanocomposite groups. Water sorption and solubility in water are critical properties with respect to long-term bonded appliances, such as orthodontic brackets and tubes. High water sorption can lead to plasticization of the polymer and a reduction in the internal-stress threshold, leading to premature failure. In this study, however, we observed reductions in both sorption and water solubility. Despite negative water-solubility values for the nanocomposite groups, this does not reflect the absence of elution, and can be reasonably ascribed to the formation of bonds to the polar groups of polymer chains, the expression of very low solubility, or early rapid desorption [36,37].

NDs mimic the properties of bulk diamond, and these properties were observed in the nanocomposite polymers. Resin-based biomedical appliances are used in intraoral environments, as brackets and removable appliances. Properties that enable these appliances to withstand a variety of mechanical insults, such as abrasion and compression, during their use are essential. Mechanical testing provides an objective way of comparing force limits and describing the elastic nature of these materials. In agreement with the findings of previous studies [28,31,38], the addition of NDs was observed to increase mechanical strength by about 31% and the elastic modulus by 12%. Among the two nanocomposite groups, the A-ND-incorporated group exhibited higher values, which is ascribable to stronger covalent linkages and the somewhat better mono-dispersed state of the A-NDs in the polymer, as evidenced by TEM [39].

The findings were further validated through the notable improvements in the moduli of resilience of the nanocomposites, which indicate an augmented capacity to absorb energy prior to deformation, as observed by SEM, where increased coarseness (A-ND >ND) was observed in the fracture surfaces. The rough surface corresponds to the energy dissipation pattern of the propagating crack, which bows around the inorganic filler particles to create step defects [40,41].

Furthermore, the surface hardness results show a similar trend, with significantly higher hardnesses observed for the nanocomposite test groups. However, in contrast to a previous study [42], the ND-incorporated group showed a slightly higher surface hardness compared to the A-ND group, although statistically they are both comparable.

The evolving trend towards smart therapeutic systems has gained attention in recent years, and, to that end, localized intricate devices such as orthodontic brackets and splints have been patented as carriers [43]. In addition, prospective applications in form of 3D-printed maxillofacial orthopedic and prosthetic appliances and scaffolds have also been proposed [44]. Thus, in the present study, ND-modified resins were evaluated for resistance to surface abrasion, simulating cleansing with a toothbrush. The ND- and A-ND-incorporated groups showed significantly improved resistances to surface wear, with minimal changes in surface roughness, which is in agreement with the findings of Ayatollahi et al. [31] for epoxy resin, where a low percentage (0.1 wt%) of added ND resulted in a consistent increase in surface hardness and a marked reduction in wear. These wear characteristics are mostly directly composed of abrasive wear and microfatigue. The loading forces in a resin composite are optimally transferred to the filler particles in the matrix [45]; hence, the properties of the nanofiller directly affect the features of the nanocomposite. In the context of the present study, we also noted that improvements in mechanical properties resulting from the addition of aminated nanodiamonds are reflected in the observed significant resistance to surface wear.

To simulate the response of the resin-based appliances in the environment of intended use, changes with hydrothermal fatigue were analyzed. A simulation, replicating the intraoral physical conditions which influence the polymer-based appliance, was assessed. In a review by Gale and Darvell, it has been suggested that the mean temperature intraorally ranges from 6 °C to 55 °C with as much as 10,000 cycles of change indicating the length of a year [19]. Under such conditions, hydrolytic aging occurs, which is believed to cause plasticization and impair the mechanical properties of the polymer by causing damage to ester bonds [46,47]. The observations of the present study were in accordance with the proposed concept and reduction in mechanical properties was noted. However, nanocomposite specimens were superior to the control group, and A-ND showed a significantly higher modulus of resilience after the hydrothermal insult.

With the ultimate aim of fabricating individualized appliances, the quality of fit is an essential aspect in 3D-printed intraoral appliances. This is of added significance when considering the complex design requirements of an orthodontic bracket. Therefore, in the present study, we used the quality quantification test to analyze printed outcomes. We printed a 3D bracket design to verify that modification of the resin polymer composition does not change the output quality. A low RMS deviation value corresponds to superior trueness, as demonstrated in earlier studies using similar methodologies that compare offset errors [20,48]. The absence of any significant differences in trueness suggest that similar printing conditions to those currently used for commercial resin provides reproducible outcomes. However, the test only compared three groups, and the printer and associated design factors also concomitantly influence the output.

The ability of A-ND to influence the properties of the UV-curable nanocomposite was successfully validated in the in-vitro environment of the present study, which highlights its promise for future development and direct clinical applications through in-vivo and longevity studies.

## 5. Conclusions

The mechanical properties of an A-ND-incorporated UV-curable resin were evaluated. The null hypothesis was rejected on the basis of the various mechanical and physical properties of the 3D-printed specimens. Significant improvements in resin properties were observed with the incorporation of 0.1 wt% A-ND, while the quality of the additively manufactured specimen was maintained. Therefore, the nanofiller-based composite is potentially a material of choice for the fabrication of 3D-printed customized biomedical appliances.

## Figures and Tables

**Figure 1 nanomaterials-10-00827-f001:**
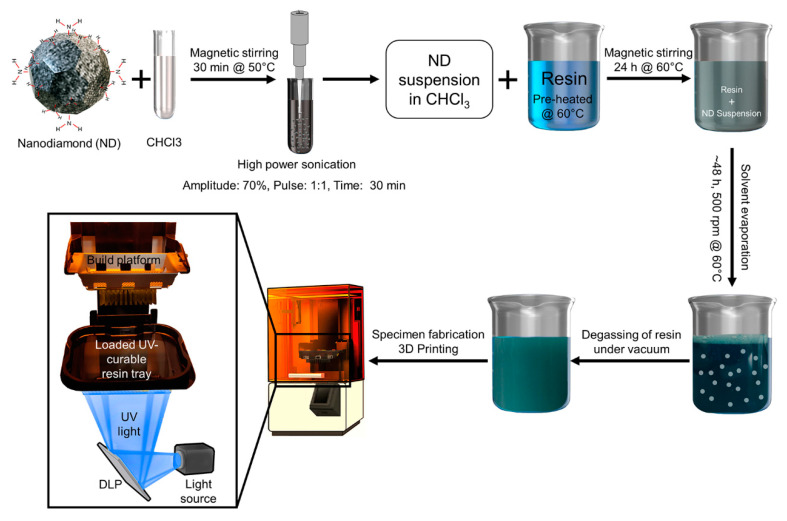
Depicting the process used to prepare and manufacture nanocomposites.

**Figure 2 nanomaterials-10-00827-f002:**
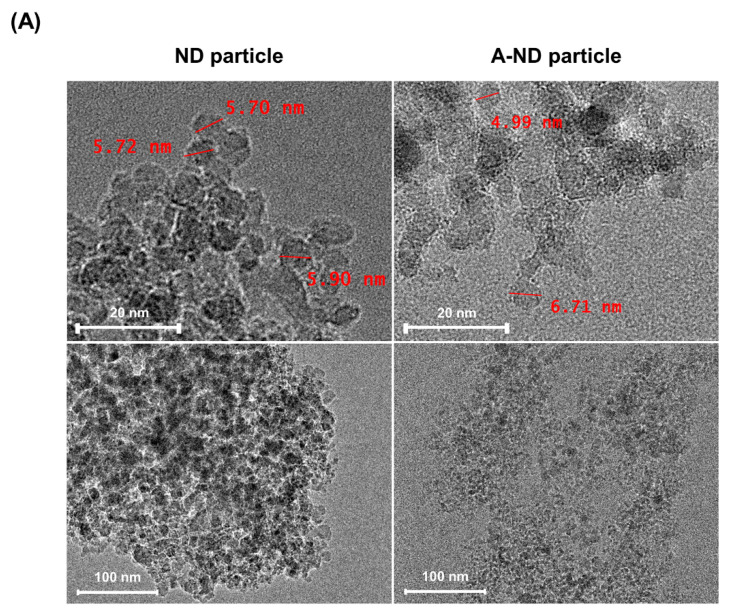
Characterization of nanodiamond particles. (**A**) TEM images of ND- and A-ND powder particles. (**B**) Histograms showing aggregate size evaluation.

**Figure 3 nanomaterials-10-00827-f003:**
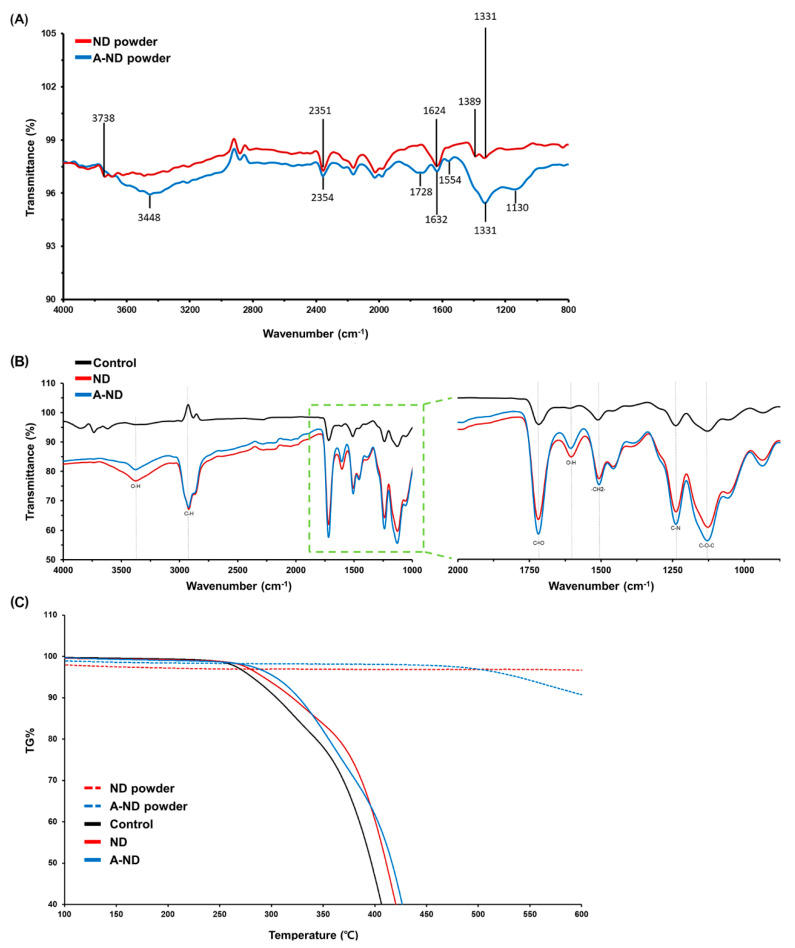
(**A**) FT-IR spectra of the ND and A-ND powders; (**B**) the control, ND- and A-ND-incorporated nanocomposites with expanded 1000–2000 cm^−1^ regions shown; (**C**) TGA profile of the ND and A-ND powders, and the control and ND- and A-ND-incorporated samples, obtained by TGA analyses performed in inert atmosphere (nitrogen).

**Figure 4 nanomaterials-10-00827-f004:**
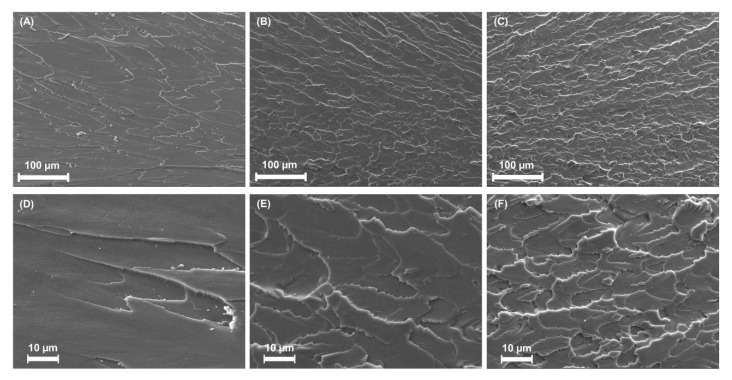
SEM images of (**A,D**) the control; (**B,E**) ND-; and (**C,F**) A-ND-incorporated nanocomposites, showing clear differences in fracture-surface patterns. (**A–C**) 250× and (**D–F**) 1000× magnification.

**Figure 5 nanomaterials-10-00827-f005:**
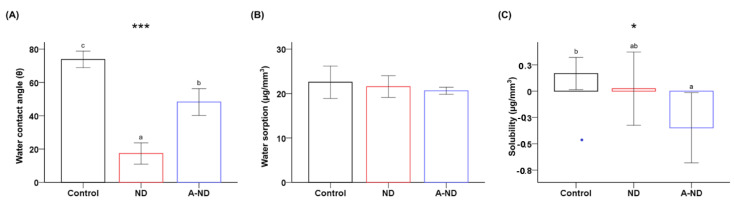
Comparing (**A**) water contact angles; (**B**) water sorptions; and (**C**) solubilities of the control, ND-, and A-ND-incorporated groups. The same lowercase letters indicate no significant difference. * *p* < 0.05, *** *p* < 0.001.

**Figure 6 nanomaterials-10-00827-f006:**
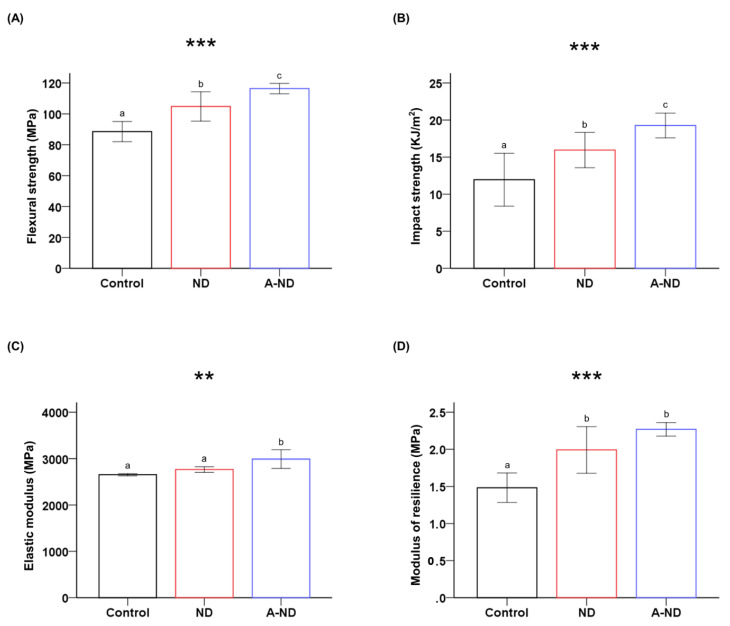
Comparing the (**A**) flexural strengths; (**B**) impact strengths; (**C**) elastic moduli; and (**D**) moduli of resilience of the various groups of samples. Different lowercase letters indicate significant differences. ** *p* < 0.01, *** *p* < 0.001.

**Figure 7 nanomaterials-10-00827-f007:**
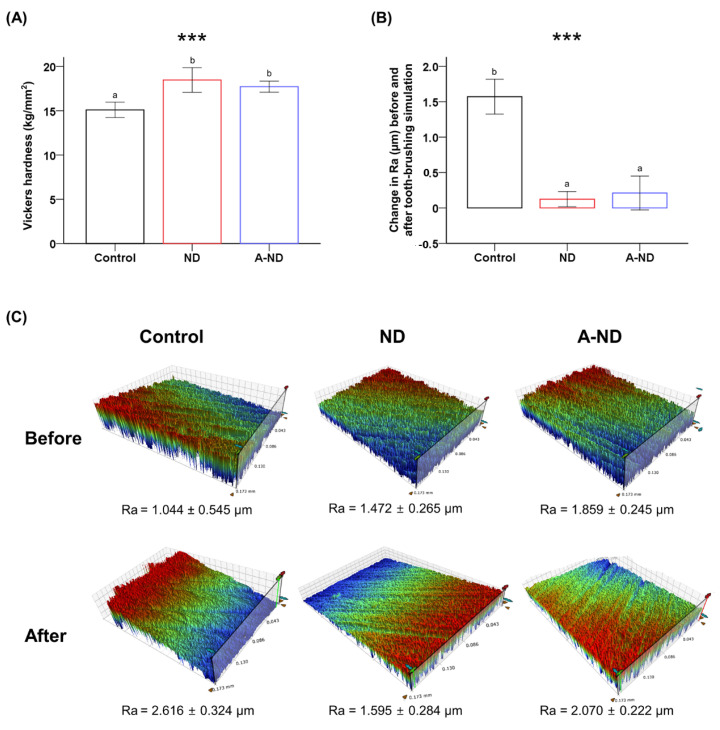
Comparing (**A**) Vickers hardness and (**B**) the mean differences in surface roughness before and after simulated toothbrushing. (**C**) Representative 3D profilometric images of the samples before and after brushing with mean roughness (Ra) values. *** *p* < 0.001 for comparison between the groups.

**Figure 8 nanomaterials-10-00827-f008:**
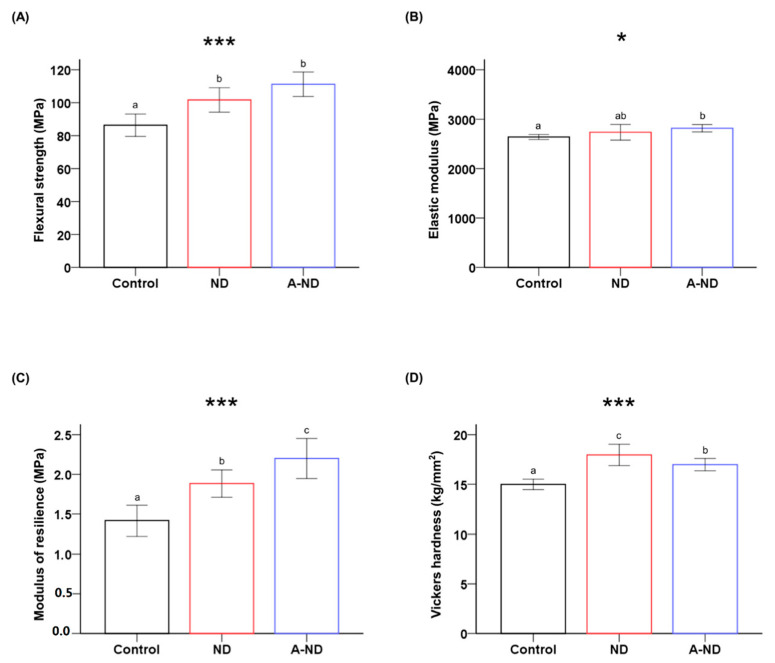
Comparing the (**A**) flexural strengths; (**B**) elastic moduli; (**C**), moduli of resilience; (**D**) Vickers hardness of the various groups of samples after exposure to hydro-thermal fatigue. Different lowercase letters indicate significant differences. * *p* < 0.01, *** *p* < 0.001.

**Figure 9 nanomaterials-10-00827-f009:**
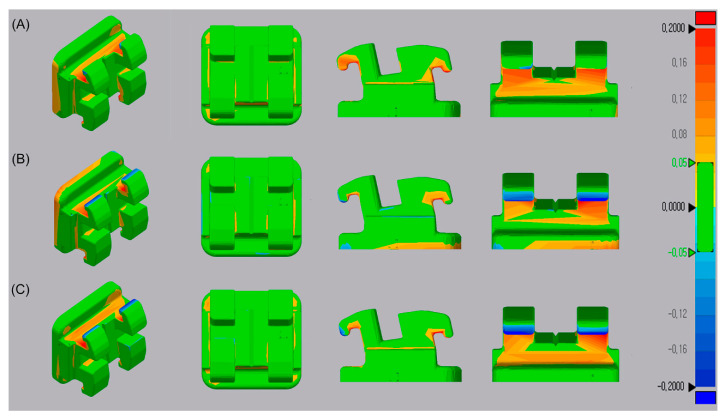
Superimposed optically scanned surfaces of 3D-printed bracket designs and the reference CAD model. (**A**) Control; (**B**) ND-incorporated; and (**C**) A-ND-incorporated.

**Table 1 nanomaterials-10-00827-t001:** Mean values of flexural strength, impact strength, elastic modulus, modulus of resilience and Vickers microhardness of the various groups and their standard deviations.

Group	Flexural Strength (MPa) ***	Impact Strength (kJ/m^2^) ***	Elastic Modulus (MPa) **	Modulus of Resilience (MPa) ***	Vickers Microhardness (kg/mm^2^) ***
Control	88.53 ± 6.53 ^a^	11.95 ± 3.56 ^a^	2654.34 ± 25.23 ^a^	1.48 ± 0.19 ^a^	15.09 ± 0.87 ^a^
ND	104.78 ± 9.49 ^b^	15.95 ± 2.38 ^b^	2766.48 ± 61.03 ^a^	1.99 ± 0.31 ^b^	18.47 ±1.38 ^b^
A-ND	116.38 ± 3.36 ^c^	19.26 ± 1.66 ^c^	2989.58 ± 200.28 ^b^	2.26 ± 0.09 ^b^	17.72 ±0.62 ^b^

The same lowercase letters indicate no significant differences in the vertical column. ** *p* < 0.01, *** *p* < 0.001.

**Table 2 nanomaterials-10-00827-t002:** Mean and standard deviation values of flexural strength, elastic modulus, modulus of resilience and Vickers microhardness for different groups after thermocycling.

Groups	Flexural Strength (MPa) ***	Elastic Modulus (MPa) *	Modulus of Resilience (MPa) ***	Vickers Microhardness (kg/mm^2^) ***
Control	86.35 ± 6.78 ^a^	2640.00 ± 50.25 ^a^	1.41 ± 0.19 ^a^	14.98 ± 0.52 ^a^
ND	101.76 ± 7.44 ^b^	2735.08 ± 158.70 ^ab^	1.88 ± 0.17 ^b^	17.96 ± 1.07 ^c^
A-ND	111.21 ± 7.43 ^b^	2816.25 ± 73.83 ^b^	2.20 ± 0.25 ^c^	16.98 ± 0.62 ^b^

Same lowercase letters indicate no significant differences in vertical column. * *p* ≤ 0.05, *** *p* ≤ 0.001 for comparison between the groups.

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
