# Peer review of "Incorporating Aminated Nanodiamonds to Improve the Mechanical Properties of 3D-Printed Resin-Based Biomedical Appliances"

_nanomaterials, 2020, doi:10.3390/nano10050827_

Round 1
Reviewer 1 Report
The paper sent for review describes the results obtained during the research on the use of nano- fillers (aminated and non-aminated nanodiamonds) in the formulation of polyacrylic resin compositions intended for 3D printing DLP method, for biomedical use. The aim of the research was to examine the impact of the content of this nanocomposite on the properties of the products obtained by printing 3D method. This work is a continuation of the earlier work of the authors on the properties of polymer / nanodiamond nanocomposites. The manuscript presented is unfortunately not of sufficiently high scientific quality, the presented works are mainly of engineering work character.
Due to the practical significance of the carried out research and the possible widespread interest in it, the manuscript can be published in Nanomaterials. So it is possible only after introducing many changes, additions and explanations to the text. Only then the scientific level of the work will be appropriate to the rank of the Nanomaterials journal. In order for the work to be scientific, it is necessary to study the effect of filler content on changes in properties (examine several different ND and A-ND contents as well as changes of interactions between the filler and the matrix). There is also no basic research on the possible impact of the filler on the resin crosslinking process. The following issues should also be resolved.
Chapter 2.1 -
Even the basic characteristics of the used nanodiamonds are lacking. Please enter the exact technical characteristics of the used nanomaterial, above all, description of the degree of amination, average particle size, agglomerate size, obtained by detonation or non-detonation method, etc. It is also necessary to accurately characterize the properties of the used acrylic resin.
Chapter 2.2 – The presented description of the method show the parameters used during the formation of the resin / nanodiamond composite. Have these parameters been obtained as a result of optimization tests? If so, how did the homogeneity of the mixture change with increasing mixing time, what is the role of higher temperature, how long did sonication last? How long after obtaining this composite, it was used in 3D printing. After what time sedimentation of the filler is observed, do the agglomeration increases with time.
Chapter 3.1 - In order to determine the degree of aggregation of nanoparticles based on the analysis of TEM images, some parameter should be introduced to assess it (e.g. agglomerate content above a certain size,% fraction content, etc.). These tests should be carried out on the basis of a sufficiently large number of observations to allow a full statistical evaluation of the obtained result.
Figure 3. – The FTIR spectra were pictured, unfortunately there is practically no comment on the obtained results. In this form, their presentation makes no sense. Please analyze the changes in spectra, comment the causes.
Chapter 3.3 – The set of tests of changes in mechanical properties lacks tensile strength tests, mainly changes in Young's modulus and maximum elongation (very valuable parameters in the mechanical assessment of the material). To fully assess the behavior of this composite, at least basic fatigue resistance tests are lacking. Tests of mechanical properties due to their intended use should be carried out at human body temperature.
The conducted discussion also does not address such important aspects as the possibility of a decrease in chemical resistance, an increase in susceptibility to surface degradation and erosion as the increases of content of added nanodiamonds.
Author Response
Q1 Chapter 2.1
Even the basic characteristics of the used nanodiamonds are lacking. Please enter the exact technical characteristics of the used nanomaterial, above all, description of the degree of amination, average particle size, agglomerate size, obtained by detonation or non-detonation method, etc. It is also necessary to accurately characterize the properties of the used acrylic resin.
A1
Thank you for the valuable suggestions. As per the reviewer’s comment, details about the materials used in terms of resin and nanofillers have been included in the 2.1. The additional tests were carried out for the characterization of the nanoparticles, based on the previous literatures, which are now included in the sections 2.3 and 3.1.Q2 Chapter 2.2
The presented description of the method show the parameters used during the formation of the resin / nanodiamond composite. Have these parameters been obtained as a result of optimization tests? If so, how did the homogeneity of the mixture change with increasing mixing time, what is the role of higher temperature, how long did sonication last? How long after obtaining this composite, it was used in 3D printing. After what time sedimentation of the filler is observed, do the agglomeration increases with time.
A2
The conditions for preparation of the ND incorporated resin was based on the previous studies by Garg et al1, Suave et al2, and Prolongo et al3. These studies conducted a detailed analysis of the different parameters such as sonication duration and power. In addition, the study by Garg et al, also elaborated the two different mixing methods with and without solvent. Hence, the optimized protocol mentioned by Garg et al, also similar to graphene based reinforcement study of Feng et al4, was adapted in the present study.
The prepared mix, after elimination of Chloroform, was subjected to degassing in vacuum (2h) and used for printing immediately, without any additional treatment.
The explanation to the same has been appended in the Discussion part of Section 4, lines 386-392, with additional referencing including the ones cited below.
References:
-
Garg, P., Singh, B. P., Kumar, G., Gupta, T., Pandey, I., Seth, R. K., Tandon, R. P., & Mathur, R. B. (2011). Effect of dispersion conditions on the mechanical properties of multi-walled carbon nanotubes based epoxy resin composites. Journal of Polymer Research, 18(6), 1397–1407. https://doi.org/10.1007/s10965-010-9544-8
-
Suave, J., Coelho, L. A. F., Amico, S. C., & Pezzin, S. H. (2009). Effect of sonication on thermo-mechanical properties of epoxy nanocomposites with carboxylated-SWNT. In Materials Science and Engineering A (Vol. 509, Issues 1–2, pp. 57–62). https://doi.org/10.1016/j.msea.2009.01.036
-
Prolongo, S. G., Burón, M., Gude, M. R., Chaos-Morán, R., Campo, M., & Ureña, A. (2008). Effects of dispersion techniques of carbon nanofibers on the thermo-physical properties of epoxy nanocomposites. In Composites Science and Technology (Vol. 68, Issue 13, pp. 2722–2730). https://doi.org/10.1016/j.compscitech.2008.05.015
-
Feng, Z., Li, Y., Hao, L., Yang, Y., Tang, T., Tang, D., & Xiong, W. (2019). Graphene-Reinforced Biodegradable Resin Composites for Stereolithographic 3D Printing of Bone Structure Scaffolds. Journal of Nanomaterials, 2019(4), 1–13. https://doi.org/10.1155/2019/9710264
Q3 Chapter 3.1
In order to determine the degree of aggregation of nanoparticles based on the analysis of TEM images, some parameter should be introduced to assess it (e.g. agglomerate content above a certain size,% fraction content, etc.). These tests should be carried out on the basis of a sufficiently large number of observations to allow a full statistical evaluation of the obtained result.
A3
The TEM imaging was focused, on qualitatively screening of the dispersion between the two different types of nano-diamond use. The statistical parameter with repeated iterations as advised by the reviewer would provide a specific image of the significance between the two. However, the TEM imaging was done in reference to previous study elucidating qualitative dispersive variation alone.
For quantitative estimation of the difference in the agglomerate formation between the two fillers, dynamic light processing was used. The results obtained from the test have been appended to Figure 2, while the details about the methods and results of the study are now included.
Q4 Figure 3
The FTIR spectra were pictured, unfortunately there is practically no comment on the obtained results. In this form, their presentation makes no sense. Please analyze the changes in spectra, comment the causes.
A4
The description of the characteristic peaks as observed and the inferences have been added to the result obtained from FTIR. References, citing similar findings and basis of interpretation have also been cited for the same.
Q5 Chapter 3.3
The set of tests of changes in mechanical properties lacks tensile strength tests, mainly changes in Young's modulus and maximum elongation (very valuable parameters in the mechanical assessment of the material). To fully assess the behavior of this composite, at least basic fatigue resistance tests are lacking. Tests of mechanical properties due to their intended use should be carried out at human body temperature.
A5
Thank you for the suggestion regarding testing based on the intended use. We now have carried out hydro-thermal fatigue test in accordance to previous studies and mechanical tests following such fatigue were carried out. Respective additions have been made in the manuscript at sections 2.9 and 3.5. We appreciate all your comments which we believe that they significantly improved our manuscript.
Reviewer 2 Report
Overall, well written and minor revision is suitable.
(1) Chemical bonding between A-ND and resin should be discussed more.
(2) Superscript and subscript are missing in many parts.
Author Response
Q1
Overall, well written and minor revision is suitable.
A1
Thank you for the encouraging comment. The suggested changes for the improvement of the manuscript have been duly noted and rectified.
Q2
Chemical bonding between A-ND and resin should be discussed more
A2
In the previous study by Mochalin et al, covalent bonding has been suggested to occur between the resin and the aminated nanodiamond fillers. Similar trends in the FTIR imaging were observed and have been elaborated.
Q3
Superscript and subscript are missing in many parts.
A3
The authors would like to thank the reviewer and apologize for the lapse in formatting. The manuscript has been verified to reflect the changes in result values and chemical formulas.
Round 2
Reviewer 1 Report
The authors made changes and supplemented the manuscript text in line with my previous suggestions. Virtually all of my comments have been taken into account and authors dispelled my previous doubts. That is why I believe that the work after changes has been made suitable for publication.